# Controlling for the species-area effect supports constrained long-term Mesozoic terrestrial vertebrate diversification

Roger A. Close[1], Roger B.J. Benson[2], Paul Upchurch[3] & Richard J. Butler[1]

Variation in the geographic spread of fossil localities strongly biases inferences about the evolution of biodiversity, due to the ubiquitous scaling of species richness with area. This obscures answers to key questions, such as how tetrapods attained their tremendous extant diversity. Here, we address this problem by applying sampling standardization methods to spatial regions of equal size, within a global Mesozoic-early Palaeogene data set of non-flying terrestrial tetrapods. We recover no significant increase in species richness between the Late Triassic and the Cretaceous/Palaeogene (K/Pg) boundary, strongly supporting bounded diversification in Mesozoic tetrapods. An abrupt tripling of richness in the earliest Palaeogene suggests that this diversity equilibrium was reset following the K/Pg extinction. Spatial heterogeneity in sampling is among the most important biases of fossil data, but has often been overlooked. Our results indicate that controlling for variance in geographic spread in the fossil record significantly impacts inferred patterns of diversity through time.

[1] School of Geography, Earth and Environmental Sciences, University of Birmingham, Edgbaston, Birmingham B15 2TT, UK. [2] Department of Earth Sciences, University of Oxford, Oxford OX1 3AN, UK. [3] Department of Earth Sciences, University College London, London WC1E 6BT, UK. Correspondence and requests for materials should be addressed to R.A.C. (email: r.a.close@bham.ac.uk).

Modern ecosystems on land are richly diverse, but there is little agreement on how this diversity accrued through geological time[1–8]. Most disagreement surrounds the degree to which ecological limits constrain patterns of diversification on continental scales[9–14]. This question has profound implications for our understanding of past, present and future biodiversity.

Competitive ecological interactions between species may constrain richness via density-dependent regulation of diversification rates: as ecospace becomes more crowded, competition for finite resources increases, slowing speciation and/or accelerating the pace of extinction ( = bounded diversification)[11,12,15]. Ecological limits predict that regional diversity should follow a logistic curve, eventually reaching a dynamic equilibrium analogous to that described by MacArthur and Wilson's[16] model of island biogeography[17,18]. If ecological limits regulate diversification rates in deep time, terrestrial diversity may have been broadly comparable to present-day levels for much of its history, or undergone long-term equilibrial periods punctuated by abrupt increases caused by major environmental perturbations or the evolutionary origins of key innovations[8,15,19,20].

The alternative to bounded diversification is unbounded or unconstrained diversification. The distinction between these two modes is an important one: if increases in terrestrial species richness have been unbounded, then more species are alive today than ever before, and ecosystems might take longer to recover from future biodiversity losses. Historically, key evidence for unbounded diversification has been drawn from both the marine and the terrestrial fossil records[2,21,22].

Time series of species richness inferred from the fossil record are a critical line of evidence for distinguishing bounded versus unbounded diversification over geological timescales[23,24]. However, inferences about macroecological and macro-evolutionary processes drawn from these data hinge on the appropriate identification, interpretation and accommodation of sampling biases that are known to pervade the fossil record[6,25,26]. Many studies have used subsampling[20,27–29] and other sample-standardization methods[30] to estimate the richness of fossil assemblages through time. These methods estimate the underlying richness of the taxon pool represented by a fossil assemblage. However, they do not correct for variation in the size of the taxon pool itself. These can result from differences in the size of the geographic sampling universe between time intervals[28]. This is important, because species-area relationships are ubiquitous[16,31], and the geographical area accessible to palaeontologists ( = palaeogeographic spread) varies dramatically among intervals of deep time[8,25,29,32–34].

However, correcting for spatial sampling biases is complicated by the interdependent nature of Earth system processes[29,35]. Relationships between the apparent richness of fossil taxa and the palaeogeographic spread over which fossils have been collected could result from either of two non-mutually-exclusive models: (1) 'record bias', in which species richness covaries with sampled area purely due to the confounding effect of uneven spatial sampling and species-area relationships; or (2) 'common cause', in which Earth system processes (for example, sea-level change, tectonic activity) ultimately determine both the sizes of individual regions available for sampling and their corresponding species richness. Previous attempts to correct for variation in palaeogeographic spread[8,32–34,36–39] would not allow the roles of these two alternative models to be distinguished, making it difficult to determine whether the corrections were appropriate (under a record bias model) or not (under a common cause model). Nevertheless, the issue can be circumvented by drawing fossil occurrences from geographic regions of equal size through

time and space before the application of richness-estimation methods.

Tetrapods, the limbed vertebrates, are highly diverse both today and in the fossil record, and have been used as a key example of unconstrained diversification in the debate about long-term diversity patterns on land[2,13,40]. Raw (that is, uncorrected) counts of fossil tetrapod taxa have been used as evidence that their diversity increased in an unbounded, exponential fashion through the Phanerozoic[2,5,41–43]. However, smaller-scale studies of diversification, in subclades of tetrapods and on regional geographic scales[7,15,44,45] provide evidence for equilibrial dynamics, mirroring the findings of most studies of global marine diversification patterns[20,24,46–48].

Recently, some of us conducted an analysis of regional diversity patterns in Mesozoic-early Palaeogene non-flying terrestrial tetrapods. This analysis corrected for uneven sampling intensity by applying a coverage-based rarefaction technique, Shareholder Quorum Subsampling (SQS)[8], to a global species-level occurrence data set. That study found a substantially smaller increase in tetrapod diversity through time than that implied by previous studies: species richness was inferred to have at most doubled over the ∼190 million year span of the Mesozoic. This provides evidence for bounded diversification, and implies a long, almost static interval of tetrapod diversification during the Mesozoic (it would be impossible to generate the large number of extant tetrapod species at constant diversification rates if their richness only doubled during half of their evolutionary history). If spatial biases are important, however, even a doubling of diversity could be an overestimate, as the palaeogeographic spread of tetrapod fossil localities within continents shows a pronounced tendency to increase through the Mesozoic[8]. Moreover, subsampled regional diversity estimates were significantly correlated with palaeogeographic spread[8] despite implementing a three-collections-per reference algorithm[49] explicitly aimed at holding the size of the geographic sampling universe more constant. This underscores the need to standardize spatial sampling.

To investigate the impact of uneven spatial sampling on temporal patterns of species richness in terrestrial fossil vertebrates, we estimated sample-standardized richness for 27,260 global tetrapod occurrences of 4,898 species in 3,323 genera while also standardizing the amount of palaeogeographic spread sampled for individual diversity estimates (Fig. 1). Drawing spatial samples of fixed extent is superior to *post hoc*, residual-based modelling approaches that have been used in the past to correct for sampling biases introduced by systematic variation in rock volume or outcrop area (for example, refs 6,50–54), as it does not impose a fixed relationship between diversity and the confounding variable. We find that controlling for variability in palaeogeographic spread eliminates even the modest increases in subsampled standing diversity over the course of the Mesozoic that are inferred when palaeogeographic spread is not standardized. An abrupt increase in diversity in the earliest Palaeogene is recovered, consistent with a resetting of equilibrial diversity levels across the Cretaceous/Palaeogene (K/Pg) mass extinction that resulted in a new dynamic equilibrium level for early Cenozoic tetrapods.

## Results

**Standardization of spatial sampling.** We present a new method that controls the confounding effect of spatial sampling variability on taxonomic diversity estimates made from fossil data (Methods section). This method was used to reconstruct patterns of species richness for Mesozoic-early Palaeogene non-flying terrestrial tetrapods[8,29]. The distribution of spatial samples is conceptually multivariate, and can be described in terms of spatial coverage,

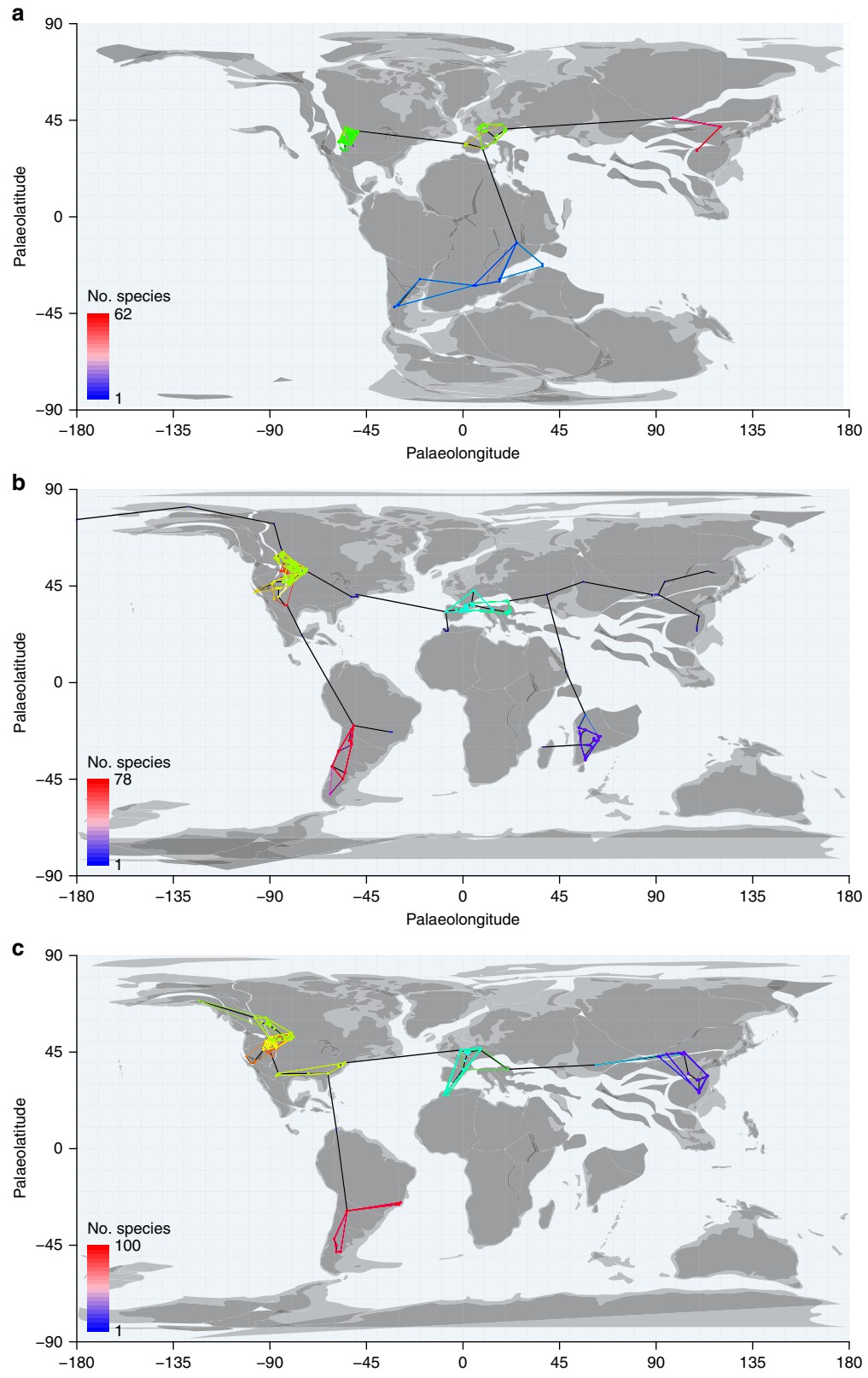

**Figure 1 | Palaeomaps of fossil occurrences through time.** Global MST shown in black and equal-area spatial subsamples enclosed by coloured convex hulls. Fossil occurrences are binned into 1° grid cells. Heatmap colours of nodes falling outside equal-area subsamples indicate number of species known from each grid cell, to demonstrate that subsamples do not omit exceptionally diverse localities. (**a**) J6 = Kimmeridgian, Tithonian (Late Jurassic); (**b**) K8 = Maastrichtian (Late Cretaceous); (**c**) Pg1 = Selandian, Thanetian (early Palaeogene). Palaeomaps drawn using shapefiles from the Scotese PALEOMAP project[80].

dispersion and total extent. Fossil localities consist of discrete, and typically unevenly distributed spatial points containing species occurrences, so for most real data sets it should not be possible to standardize over all aspects of spatial distribution simultaneously. Nevertheless, our goal here was to reduce the variance in spatial spread among sets of samples and investigate the effect of doing so on variance in diversity through time and among regions. Arguably, this could be achieved by using any of a number of different measures of spatial spread.

In practice, we quantified palaeogeographic spread using summed minimum-spanning tree (MST[55]) length, the minimum total distance of segments capable of connecting the palaeocoordinates of all fossil localities when binned to 1° grid cells. Our analyses suggest that this represents a good compromise among the distinct aspects of the spatial distribution of point localities (discussed further in the Methods section and Supplementary Methods), that is tightly correlated with other commonly used spatial metrics, including convex-hull area (Pearson's $r = 0.91$), maximum great-circle distance ($r = 0.97$), standard distance deviation ($r = 0.91$) and grid-cell occupation ($r = 0.71$; grid size = 2°; all other spatial measures correlate less well with grid-cell occupation; see Supplementary Methods and Supplementary Figs 1,2 and 7). Summed MST length therefore incorporates a combination of spatial signals that describe the overall size of the geographic sampling universe. Standardization of the palaeogeographic spread of fossil localities within continental regions was achieved by stochastically subsampling sets of localities to a summed MST length of ∼3,200 km. Full details of our spatial subsampling algorithms are presented in the Methods section and Supplementary Methods.

Following spatial standardization, two sampling standardization methods were then used to estimate species richness of spatial subsamples: (1) equal-coverage subsampling ('shareholder quorum' subsampling[20,29,56,57], and (2) the 'true richness estimated using a Poisson sampling' (TRiPS) method[30].

**Raw richness estimates**. Controlling for variability in spatial sampling through time barely alters the positive trend in raw ( = face-value, empirical, uncorrected or observed) counts of Mesozoic in-bin species richness through time, when compared to regional or variable-spread species counts (Fig. 2). General linear models (GLMs) of raw diversity against time (bins Tr4-K8; log-link function) are significant at alpha of 0.05 for unstandardized area ($P = 0.0016$; slope = −0.007; Table 1; as previously shown[8]). This results from the presence of influential data points in the later Mesozoic that meet our minimum quality criterion of being associated with at least 20 publications but which have low raw diversity (for example, North America K4 or South America K6). Standardizing area results in a GLM with a slightly steeper positive slope and high statistical significance ($P = 0.0028$; slope = −0.0071). This occurs because many influential data points representing poorly-sampled regions/intervals with low richness are removed by this procedure—particularly from the latter half of the Cretaceous. Spatially standardized samples could not be obtained for these regions because fossils have not been collected over a sufficient palaeogeographic spread. Much of the apparent increase through time results from the extensive sampling of North American sites in the latter part of the Cretaceous (bins K7–K8, or Campanian–Maastrichtian). However, the magnitude of increase in raw richness for these data points relative to well-sampled earlier intervals, such as Tr4 and J6 in North America, is reduced after standardizing geographic spread, and the raw richness of K7 in North America becomes more comparable to that for K7 in Asia. The reduction of raw richness of North America in K7 and K8 also amplifies the apparent increase in raw richness across the K/Pg

boundary. Raw richness in Pg0/Pg1 in North America is nearly twice that of K7/K8, whereas Pg2 is approximately four times as rich.

The Palaeogene tetrapod fossil record of North America is approximately one order of magnitude better sampled than the rest of the world, as measured by counts of fossil occurrences in the Paleobiology Database (PaleoDB). Over 5,000 occurrences derive from the Ypresian of North America, but only ∼200 are from Europe. This view is supported by three measures of sampling effort within equal-spread samples through time: the coverage estimator Good's $u$ (the proportion of non-singleton taxa, which increases with sampling effort as additional occurrences of known taxa are discovered); dominance (the relative frequency of the most common taxon, which tends to decrease with greater sampling effort); and the maximum-likelihood estimates of sampling intensity reported by TRiPS (Fig. 3).

**Subsampled richness estimates (SQS)**. Standardizing the level of palaeogeographic spread while simultaneously correcting for uneven sampling intensity using equal-coverage subsampling ( = 'shareholder quorum' subsampling, or SQS) largely eliminates any Mesozoic increase in standing diversity (Fig. 2c,d). Subsampled diversity estimates for continental regions, with unstandardized palaeogeographic spread, show a weak but statistically significant positive trend of increasing richness through time (GLM $P = 0.0258$; slope = −0.0038; Table 1). In contrast, the trend for subsampled diversity with standardized palaeogeographic spread is very weak and is not significantly different from zero (GLM $P = 0.1207$; slope = −0.0019). In fact, the estimated slope implies a diversity increase of only 44% over 186 million years from the start of the Triassic to the end of the Cretaceous (or 32% for the GLM for Tr1-K8). Standardizing geographic spread decreases the relative subsampled richness of the most exceptionally well-sampled points in time and space, notably North America in K7–K8 (Campanian–Maastrichtian; for example, Dinosaur Park and Hell Creek formations). After standardization of geographic spread, the estimated species richness of each of these data points is comparable to those for North American samples from Tr4 and J6. The regional subsampled diversity trajectory within North America is thus flat for much of the Mesozoic, and standardized diversity estimates for other regions such as Europe and Africa tend to confirm this (Fig. 2c,d).

**TRiPS richness estimates**. We also reconstructed diversity patterns using an alternative, and recently proposed[30], maximum-likelihood-based richness estimator, TRiPS. For regional (unstandardized) samples of palaeogeographic spread, TRiPS recovers a pattern very similar to that observed for raw richness, although confidence intervals reveal high levels of uncertainty for many richness estimates (Fig. 2e). GLMs demonstrate a statistically significant positive trend through the Mesozoic ($P = 0.0293$, slope = −0.0061). Standardizing palaeogeographic spread before estimating richness with TRiPS recovers a non-significant Mesozoic diversity trend (GLM $P = 0.2256$, slope = −0.0039), but with substantially greater scatter of individual data points than for SQS (Fig. 2f). Overall patterns of TRiPS richness estimates for area-standardized data correlate strongly with those obtained from SQS (Pearson's $r = 0.86$; by contrast, TRiPS richness estimates correlate with raw richness with a coefficient of 0.84, while SQS subsampled diversity correlates with raw diversity with a coefficient of 0.91). Furthermore, TRiPS finds an abrupt increase across the K/Pg, similar to that obtained using SQS subsampling, and corresponding to an approximate doubling in richness.

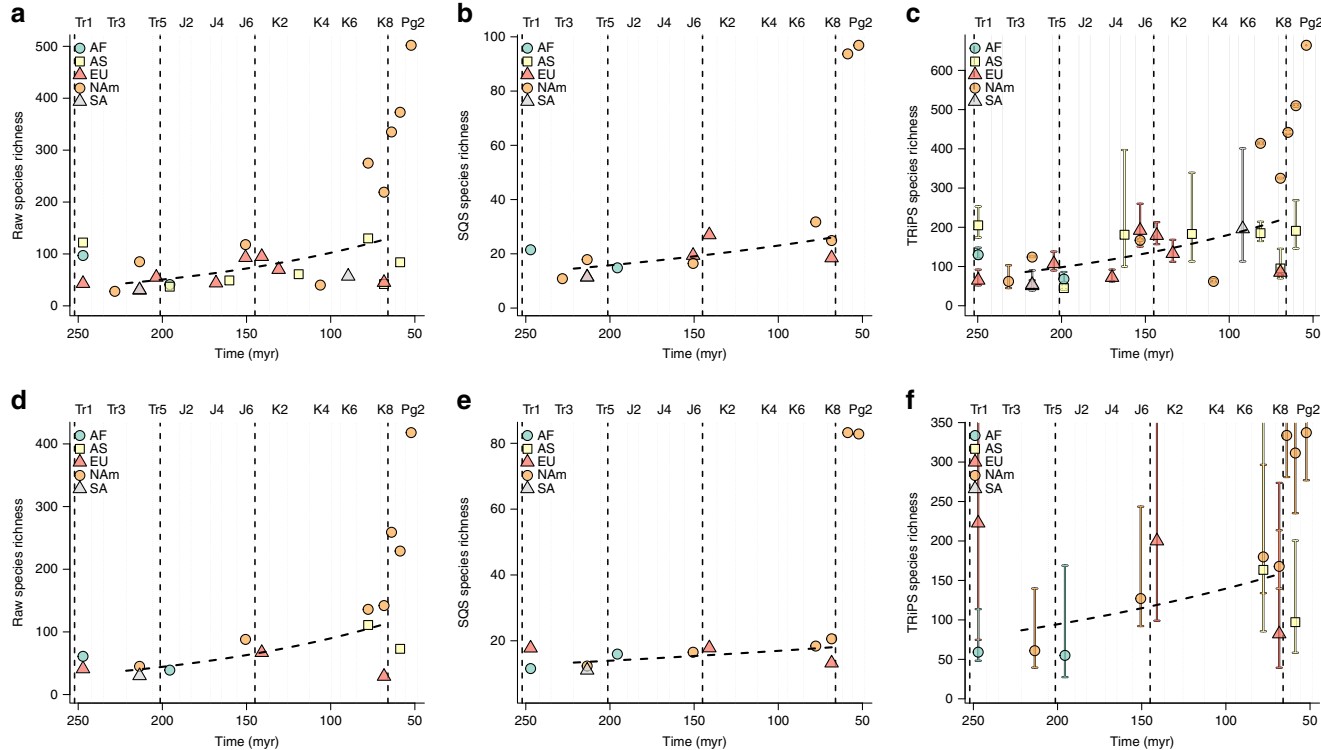

**Figure 2 | Effect of spatial and occurrence-level sampling standardization on richness.** Patterns of species richness through time for Mesozoic-early Palaeogene non-marine, non-flying tetrapods, contrasting results obtained from unstandardized (regional-level) and standardized (equal-area) levels of palaeogeographic spread. (**a**) Unstandardized-area raw diversity (slope $= -0.007$; $P = 0.0016$, degrees of freedom $= 19$); (**b**) unstandardized-area SQS subsampled diversity (slope $= -0.0038$; $P = 0.0258$, DF $= 9$); (**c**) unstandardized-area TRiPS estimated richness (slope $= -0.007$; $P = 0.0016$, DF $= 19$); (**d**) standardized-area raw diversity (slope $= -0.0071$; $P = 0.0028$, DF $= 8$); (**e**) standardized-area SQS subsampled diversity (slope $= -0.0019$; $P = 0.1207$, DF $= 7$); (**f**) standardized-area TRiPS estimated richness (slope $= -0.0039$; p $= 0.2256$, DF $= 7$; error bars represent 95% maximum-likelihood estimate confidence intervals). Only data points associated with at least 20 references are shown. Trend lines represent results of GLMs summarized in Table 1 for bins Tr4-K8. Combinations of colours and symbols distinguish continental regions, which are defined in Supplementary Table 1.

**Table 1 | Relationship between time and species richness for unstandardized (regional) and standardized levels of palaeogeographic spread, for intervals Tr1-K8 and Tr4-K8.**

|  | Slope | Standard error (slope) | *P* (slope) | Bonferroni-corrected *P* (slope) | DF | Ln increase | Percent increase |
|---|---|---|---|---|---|---|---|
| Unstandardized area (raw) Tr1-K8 | $-0.004$ | 0.0019 | 0.0317 | 0.3807 | 23 | 0.7537 | 112 |
| Unstandardized area (raw) Tr4-K8 | $-0.007$ | 0.0022 | 0.0016 | 0.0196 | 19 | 1.3058 | 269 |
| Standardized area (raw) Tr1-K8 | $-0.0052$ | 0.0018 | 0.0035 | 0.0416 | 10 | 0.977 | 166 |
| Standardized area (raw) Tr4-K8 | $-0.0071$ | 0.0024 | 0.0028 | 0.033 | 8 | 1.3177 | 273 |
| Unstandardized area (SQS) Tr1-K8 | $-0.0033$ | 0.0012 | 0.0223 | 0.2677 | 11 | 0.6193 | 86 |
| Unstandardized area (SQS) Tr4-K8 | $-0.0038$ | 0.0014 | 0.0258 | 0.3096 | 9 | 0.7056 | 103 |
| Standardized area (SQS) Tr1-K8 | $-0.0015$ | 9.00E-04 | 0.1292 | 1 | 9 | 0.2781 | 32 |
| Standardized area (SQS) Tr4-K8 | $-0.0019$ | 0.0011 | 0.1207 | 1 | 7 | 0.3624 | 44 |
| Unstandardized area (TRiPS) Tr1-K8 | $-0.005$ | 0.002 | 0.0192 | 0.2304 | 23 | 0.938 | 155 |
| Unstandardized area (TRiPS) Tr4-K8 | $-0.0061$ | 0.0026 | 0.0293 | 0.3512 | 19 | 1.141 | 213 |
| Standardized area (TRiPS) Tr1-K8 | $-0.0018$ | 0.0023 | 0.4486 | 1 | 9 | 0.3418 | 41 |
| Standardized area (TRiPS) Tr4-K8 | $-0.0039$ | 0.0029 | 0.2256 | 1 | 7 | 0.7284 | 107 |

Species richness quantified as raw (face-value) counts, SQS subsampled diversity (quorum $= 0.4$) and TRiPS estimated richness. Relationships for raw diversity represent GLMs assuming a negative binomial error distribution and a log-link function; relationships for SQS and TRiPS assuming a Gaussian error distribution and log-link function.

**Correction for multiple comparisons**. Bonferroni corrections for multiple comparisons modify our threshold for statistical significance to 0.0042, following which our GLMs provide statistically significant evidence of increasing diversity through time only for three analyses of raw diversity counts (Table 1: unstandardized area Tr4-K8, standardized area Tr1-K8 and standardized area Tr4-K8). However, this correction is not conservative in the context of our analyses, which test the hypothesis that diversity changed little through the Mesozoic.

Furthermore, corrections for multiple comparisons are now widely disfavoured because they reduce statistical power[58], and we do not consider them to be appropriate.

**Mesozoic latitudinal biodiversity gradients**. Our analyses provide insights into the evidence available to evaluate the latitudinal biodiversity gradient during the Mesozoic, because our spatially standardized palaeogeographic samples

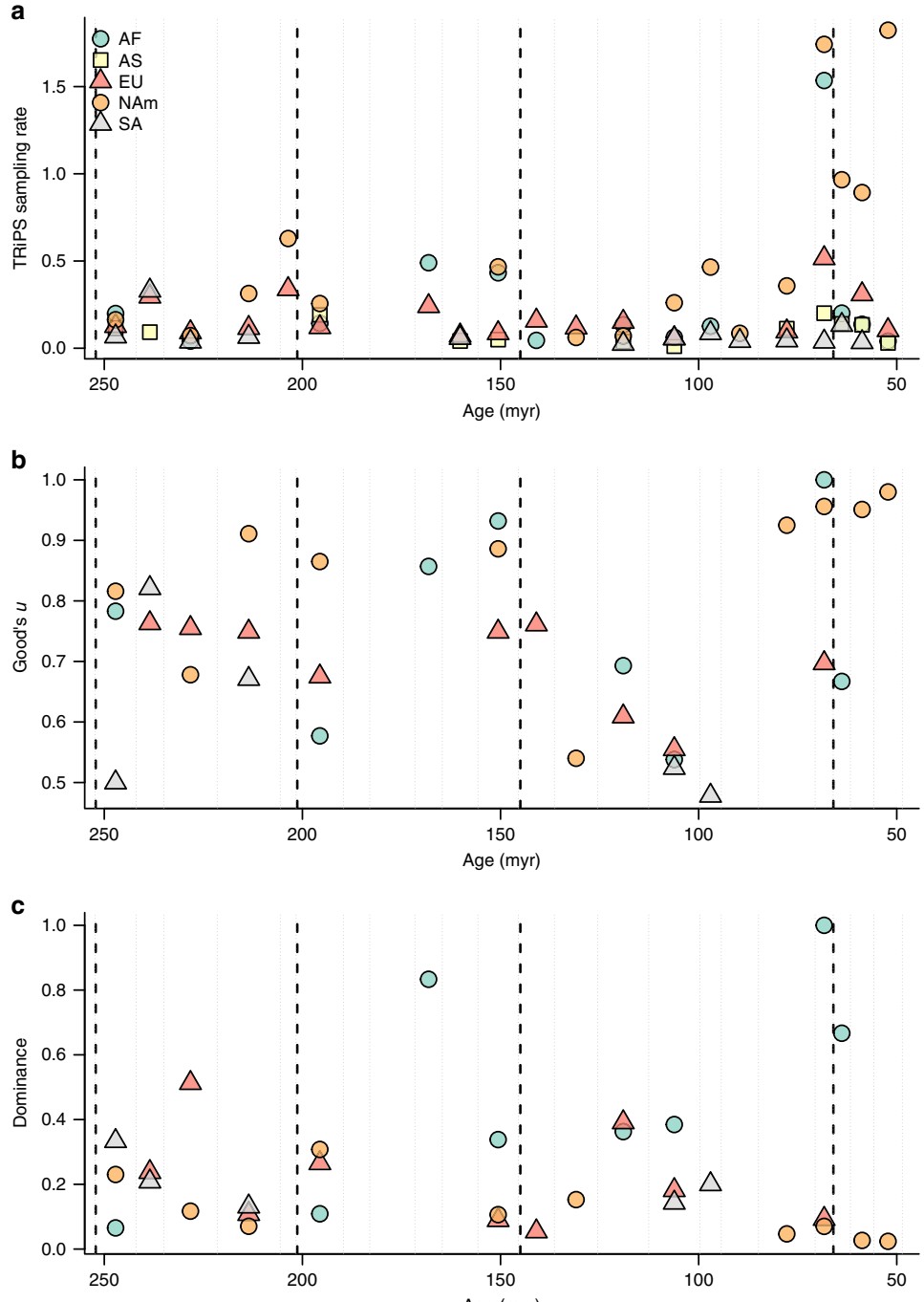

**Figure 3 | Sampling metrics through time for fossil localities at regional level.** (**a**) Number of occurrences; (**b**) Good's *u*; and (**c**) dominance. Combinations of colours and symbols distinguish continental regions, which are defined in Supplementary Table 1.

indicate which spatial/temporal regions can provide informative estimates of diversity (using SQS or TRiPS) at a spatial resolution of ~3,200 km MST length (Fig. 4). The distribution of pooled Mesozoic samples (Fig. 4b,d; see also Supplementary Fig. 3) indicates that almost all informative spatial regions derive from a narrow range of latitudes between 30° and 60°. Very few diversity estimates are available from lower- and higher-latitude fossil localities[59], as also appears to be the case for the Phanerozoic marine invertebrate record[33]. The situation is even more acute when latitudinal patterns of diversity are assessed for single time slices, because only a few intervals (Tr1, Tr4, K7 and K8) provide diversity estimates from more than one

palaeolatitude. In fact, subsampled richness estimates are only sufficient to estimate diversity at low palaeolatitudes in two time intervals (Tr1 and Tr4 in North America using SQS; and only Tr1 using TRiPS). These data points provide weak evidence for slightly lower diversity at higher latitudes compared to low latitudes. Higher diversity at lower latitudes has been reported for Triassic pseudosuchians (the total-group of Crocodylia)[60].

Although tetrapod fossils have been collected from low-latitude localities during other intervals/regions, these do not provide adequate information on species richness, because each has either insufficient geographic spread or contains too high a proportion

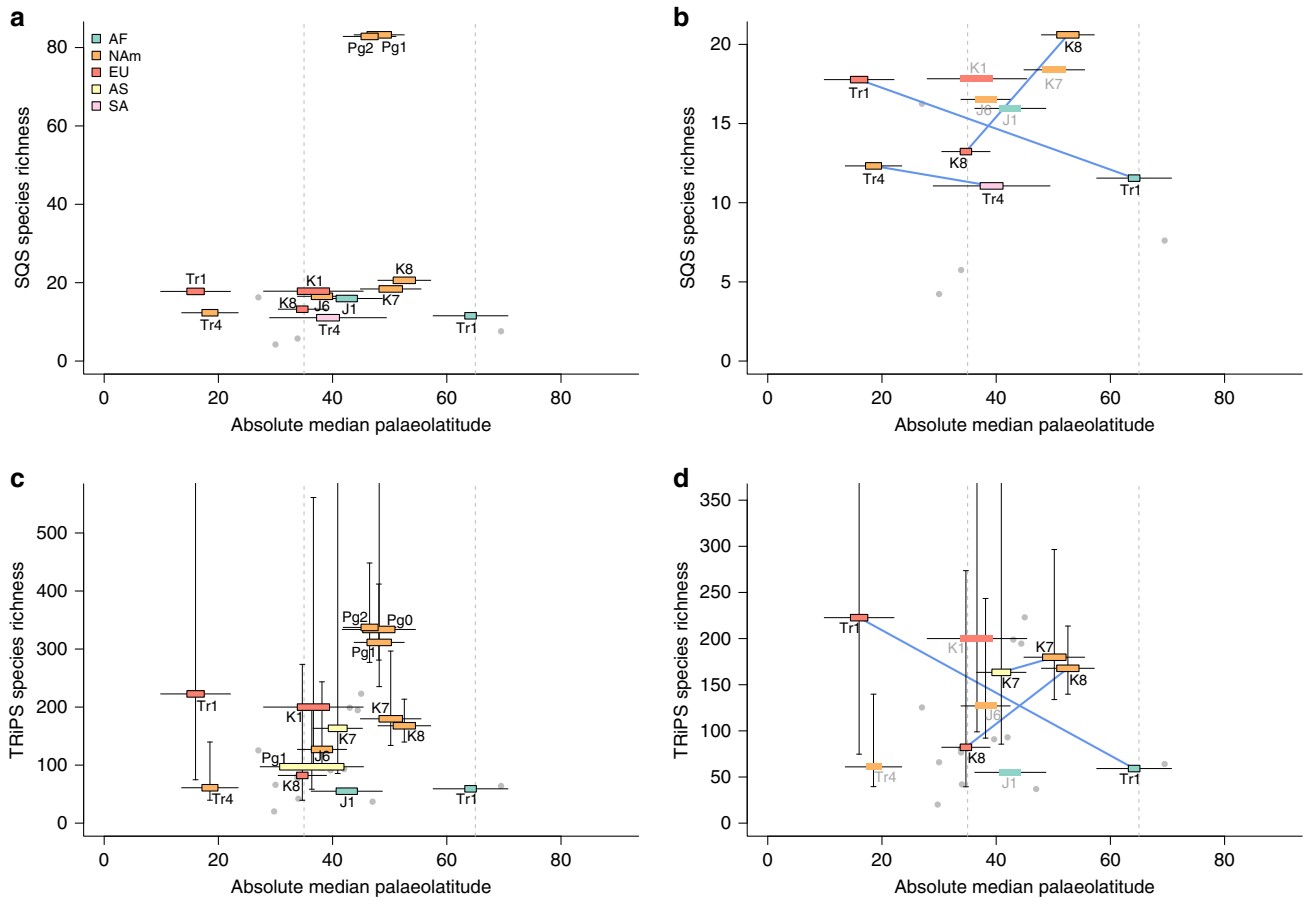

**Figure 4 | Relationship between absolute palaeolatitude and spatially standardized data.** Relationship between absolute palaeolatitude and species richness or sampling intensity for Mesozoic-early Palaeogene non-marine, non-flying tetrapods, estimated from spatially standardized samples of fossil localities. (**a**) SQS subsampled diversity for Mesozoic and early Paleogene; (**b**) SQS subsampled diversity for Mesozoic only; (**c**) TRiPS estimated richness for Mesozoic and early Paleogene; (**d**) TRiPS estimated richness for Mesozoic only. Dashed lines delimit palaeotemperate latitudes (35–65° palaeolatitude). Thick error bars represent palaeolatitudinal interquartile range of spatial subsample; thin lines represent total latitudinal range. Dotted blue lines connect regions within time intervals. Data points associated with fewer than 20 references shown in grey. Colours distinguish continental regions, which are defined in Supplementary Table 1.

of singleton occurrences to be subsampled to our target quorum level (Supplementary Fig. 3b). The apparent temperate-latitude peak in Mesozoic tetrapod diversity reported by ref. 61 may therefore be an artefact of intensive sampling of the terrestrial fossil record of palaeotemperate regions.

## Discussion

Standardizing richness estimates to uniform spatial regions demonstrates that: (1) the regional diversity of non-flying terrestrial tetrapods was essentially flat for most of the Mesozoic, and that large increases in raw species counts in the well-sampled North American record disappear entirely after spatial standardization; and (2) regional standing diversity recovered exceptionally rapidly from the K/Pg extinction. By Pg1 (Selandian–Thanetian) tetrapod diversity already exceeded pre-extinction levels by a factor of approximately three to four (although this does not preclude a transient drop in diversity immediately after the K/Pg that was too brief to be captured by the temporal resolution of our time bins). These patterns strongly support bounded diversification in the Mesozoic, followed by explosive diversification after the mass extinction, during which terrestrial tetrapod diversity increased rapidly to a new and substantially higher equilibrium level (consistent with the notion of a time-varying equilibrial diversity value[14,15,18]).

Despite dramatic variation in spatial sampling between time intervals and geographic regions, most palaeodiversity studies to date have included all known fossil data (that is, 'global' fossil data) when tabulating in-bin taxon counts or estimating subsampled diversity. Results are implicitly presented as estimates of gamma diversity (either global, continental or regional-scale), but in fact represent arbitrary points on a species-area curve, and may fall substantially below the desired spatial scale. This approach will yield reliable estimates of gamma diversity only when spatial sampling is uniformly high, or when it closely tracks changes in habitable area (which may vary due to Earth system processes, most notably continental flooding induced by sea-level change). Sample-standardization tools such as SQS and TRiPS cannot correct for systematic differences in the geographic scope of the sampling universe introduced by such unequal comparisons. In summary, analyses of fossil data that sum regional estimates to produce apparent 'global' diversity curves suffer from enormous variation in palaeogeographic spread through time, and are best avoided[8].

Standardizing the palaeogeographic spread of fossil localities before estimating taxon richness reduces the confounding effect of variable spatial sampling through time and between geographic regions. The resulting estimates may not equate directly with gamma or alpha (local/community-level) diversity. However, they do represent approximately comparable points on the

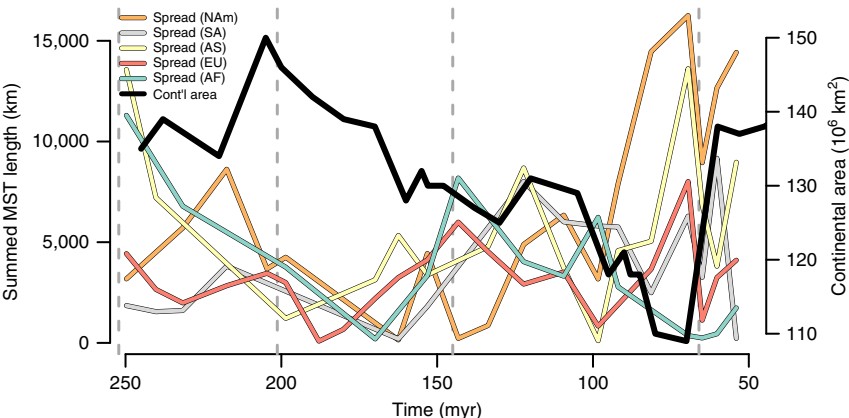

**Figure 5 | Regional palaeogeographic spread and continental land area.** Relationship between regional-level palaeogeographic spread (all data points) as quantified by summed MST length and continental land area through time. Continental land-area values derived from ref. 81. Colours distinguish continental regions, which are defined in Supplementary Table 1.

species-area curve, thus permitting fair comparisons of species richness, and how it varies through time and space. Our analyses were performed on a global fossil occurrence data set. Nevertheless, our geographic standardization procedure provides diversity estimates at regional (sub-continental) scales, which are more appropriate than global scales for studying phenomena such as diversity-dependence. Although regional and global diversities must be linked, regional diversities may sum to global diversities differently as continents fragment over geological time, as occurred in the Mesozoic[62]. Our evidence for essentially flat regional diversity patterns does not therefore preclude an overall increase in global diversity through time.

Here, and elsewhere[8], we find a strong correlation between the geographic spread of fossil localities and both raw and subsampled estimates of tetrapod species richness. This closely mirrors patterns documented for marine invertebrates[25,32,33]. Temporal variability in spatial sampling may prove to be intimately linked to macrostratigraphic biases such as the area or volume of rock available for sampling, which have long been suspected to control raw taxon counts observed in the fossil record[25,50,51,63,64]. Such rock-record megabiases are profound: terrestrial outcrops of late Carboniferous to Pliocene age on average represent <0.5% of the original landmass area, and nearly half of known terrestrial outcrops represent rocks of Cretaceous and Miocene age[6]. Raw genus diversity for all terrestrial organisms correlates strongly with terrestrial outcrop area through the Phanerozoic[6], and outcrop area strongly predicts taxon counts, as well as counts of both fossil collections and geological formations[65]. Some of these observations were originally made by Raup[25] in pioneering research into biases on fossil record estimates of biodiversity through time. They suggest that changes in rock volume or outcrop area from each time interval could be driving changes in observed diversity, biasing the fossil record, and obscuring actual patterns of ancient biodiversity.

It is unlikely that the relationship between outcrop area and fossil taxon counts result from a common cause mechanism on land. Under the 'common cause' model, the correspondence between outcrop area and fossil taxon counts results from a third variable, such as sea level, that jointly drives both factors. This differs from the 'rock-record bias' model, in which the correspondence is due to direct causation. However, fluctuations in spatial sampling do not track trends in actual habitable area. This was demonstrated by Wall et al.[6], who showed that changes in outcrop area are not significantly correlated with changes in habitable area through time in the terrestrial realm[6]. Our data

also shows no correlation between the palaeogeographic spread of fossil localities and original landmass area (Fig. 5; see also Supplementary Fig. 8), and are therefore inconsistent with a common cause model for the terrestrial fossil record. Both results favour a record bias explanation, and underscore the need to directly account for variable spatial sampling when reconstructing diversity patterns in the fossil record (*contra* refs 4,66). Our equal-spread subsampled diversity estimates demonstrate that correcting for spatial biases can yield flatter diversity trajectories relative to uncorrected data. However, we anticipate that studies of diversity in deep time will increasingly focus on quantifying species-area relationships[32,36–39,67]—which encode information about patterns of alpha, beta and gamma diversity—and how they vary through time and space. This approach will provide rich new insights about the history of biodiversity on our planet.

## Methods

**Preparation of fossil occurrence data.** To enable direct comparison with earlier work, we used the Triassic–early Palaeogene (Ypresian) tetrapod fossil occurrence data analysed by Benson et al.[8]. This data set was downloaded from the Paleobiology Database (PaleoDB) via the Fossilworks interface (http://www.fossilworks.org) on 22 January 2015. The data represent an estimated 6,520 h of work, of which the main contributors (in order of effort) were M. T. Carrano, J. Alroy, R. J. Butler, P. D. Mannion, R. B. J. Benson, A. M. Rees, W. Kiessling, M. E. Clapham, F. T. Fürsich, M. Aberhan and M. D. Uhen. Data preparation and analysis were performed in R 3.2.3 (R Core Team 2013). Data were cleaned by removing the following: occurrences that were generically indeterminate; wastebasket taxa; marine tetrapods; and both oo- and ichnotaxa (eggs and footprints). Occurrences with soft-tissue preservation were excluded, as spatiotemporally restricted modes of preservation can bias coverage-based subsampling methods[29]. Flying taxa (pterosaurs, birds and bats) were likewise excluded, as their fossil records are dominated by these exceptional modes of preservation[68,69].

**Time intervals.** We restricted our analyses to occurrences dating from the start of the Triassic to the end of the Ypresian, an interval for which records of non-marine, non-flying tetrapods in the PaleoDB were recently and comprehensively vetted[8]. Whenever possible, we sought to use time bins of approximately equal duration (~9 myr; Supplementary Table 1). There is no significant trend in-bin durations (Supplementary Fig. 4). Bins Tr2, Tr5, J2–J5 and K2–K6 were excluded from the equal-spread analyses as regional-level, variable-spread subsampled diversity estimates could not be obtained for these intervals.

**Subsampled diversity estimates.** The ubiquity of sampling biases in the fossil record largely prohibits literal interpretation of raw in-bin taxon counts. Sampling standardization is therefore necessary to correct for uneven sampling intensity. We used SQS[20,29,70] to subsample at the level of occurrences, an approach which, despite recent criticism[71], has been shown to perform extremely well when additions and subtractions to the underlying sampling pool (for example, time-bin or region) are random[64], and a range of modifications have been implemented in

Alroy's SQS Perl script version 4.3 that address violations of this fundamental assumption[20,23].

In contrast to classical rarefaction, which draws equal (but not necessarily fair) subsamples of each sampling pool, and is known to flatten diversity curves by underestimating the diversity in richer pools, SQS subsamples fairly, by drawing occurrences that represent a fixed portion or coverage (sum of proportional frequencies of taxa in each sample) of the underlying species-abundance distribution, determined by the quorum level. To sample more intensively in less well-sampled intervals, the target quorum level is modified to estimate coverage of the true, underlying sampling pool using Good's $u$ (proportion of non-singleton taxa).

SQS is a non-parametric approach that makes fewer assumptions about sampling distributions than TRiPS, a recent parametric approach that calculates maximum-likelihood estimates of underlying richness by modelling fossil sampling as a Poisson process[30]. For comparative purposes, however, results were also calculated using TRiPS, both for variable and equal levels of palaeogeographic spread.

Due to the occurrence-level structure of the PaleoDB, we used occurrence-based subsampling, which defines singletons as taxa found only in one collection. In each subsampling trial, all occurrences within each collection were drawn. Only occurrences falling entirely within a bin were used to calculate subsampled diversity for that bin. A quorum level of 0.4 was used, which adequately reflects results from higher quorum levels while permitting results for most time bins (the maximum quorum level that can be achieved for each bin is equal to Good's $u$ for that interval).

**Standardizing palaeogeographic spread.** Geographic sampling in fossil occurrence data sets has been quantified in a variety of ways. Commonly used metrics include the total area enclosed by a convex hull defined by the outermost spatial points (for example by refs 32,36,37,67,72), maximum great-circle distance (the standard metric for studies that seek to estimate geographic range-sizes for taxa from fossil occurrence data; for example, by refs 73–75), mean or median pairwise great-circle distances[29,72] and counts of grid cells from which fossil occurrences have been found (for example, refs 23,29,33,74–78). To the best of our knowledge, no study has rigorously evaluated differences in the behaviour or performance of these different metrics and, with certain exceptions (for example, ref. 29), studies have rarely attempted to justify their choice of metric.

Fossil localities consist of point-pattern data that are aggregated, to varying degrees, over a wide range of spatial scales. Summarizing information about their distribution using single univariate metrics is therefore challenging: should we be principally concerned with the total extent, dispersion, density or completeness of coverage, or clustering of points? Commonly used spatial sampling metrics each emphasize different components of the distribution of fossil localities; some emphasize ranges, others dispersion or density of coverage, and differ in their sensitivity to outliers and sampling intensity (see Supplementary Methods for more detailed discussion). In fact, it is not possible to summarize all desirable information about geographic coverage using a single univariate statistic; nor is it usually possible to standardize spatial samples over all distributional aspects simultaneously.

For this reason, we chose to use an alternative measure of palaeogeographic spread that represents a good compromise between commonly used metrics[8,29]: summed MST length. MSTs (the minimum length of segments that can connect a set of points) have been used for decades to cluster various types of data based on pairwise distances[55]. Summed MST length was first proposed by ref. 29 as a measure of the palaeogeographic spread of fossil localities that simultaneously captured several key components of spatial sampling, including spatial coverage (commonly measured as the number of grid cells within a focal region that have actually yielded fossils), dispersion (as measured by average pairwise distances or standard distance) and total extent (as measured by maximum great-circle distance). This is important, as our metric should be an informative proxy for the size of the geographic sampling universe, which plays a major role in dictating the size of the underlying taxon pool available for estimating diversity in deep time. A closely related metric, the minimum total path-length between point localities, has also been applied to point-source ecological census data in a macroecological context[79]. Supplementary Fig. 1 (see also Supplementary Fig. 2) demonstrates that summed MST length correlates well with convex-hull area (Pearson's $r = 0.91$), maximum great-circle distance ($r = 0.97$) and the spatial equivalent of the standard deviation, standard distance ($r = 0.91$); it also exhibits a tighter correlation with the number of occupied grid cells ($r = 0.71$) than any other metric. Summed MST length thus represents a good compromise between other metrics, which captures a combined signal of spatial coverage, dispersion and total extent. MSTs are also algorithmically advantageous for constructing spatial samples using point data.

We obtained samples of collections representing fixed levels of palaeogeographic spread by constructing and subsampling geographic MSTs based on the palaeocoordinates of fossil localities ($=$ collections) in Fossilworks. These were calculated using rotations obtained from the Scotese PALEOMAP Project (http://www.scotese.com)[80]. The measure of geographic spread obtained from an MST may be partly sensitive to the number of sites sampled. To a certain extent, this is desirable, as it captures a signal of the coverage of localities within the study region and accounts for the closer correlation with counts of occupied grid cells.

However, to reduce the impact that numerous densely clustered localities might have on summed MST length, we followed Alroy[29] in binning the raw palaeocoordinates into approximately equal-area grid cells of 1° palaeolatitude/palaeolongitude (corresponding to ∼111 km), a procedure that removes the summed contributions of small-scale inter-locality distances. Although grid cells formed by equidistant lines of latitude and longitude do not result in perfectly uniform cell areas along latitudinal gradients (although often used; for example, refs 20,23,29,76,77), the vast majority of collections lie at palaeotemperate latitudes, and thus the procedure has a very limited impact on our results compared to the absolute total length of each MST.

For each interval, we calculated a global MST for all sampled grid cells (Supplementary Fig. 5). To obtain standardized spatial samples of fossil localities, long branches (representing intercontinental and interregional connections) were removed from the global MST by (1) iteratively removing the longest branch, (2) calculating the individual summed MST lengths of the remaining subtrees and (3) dropping any subtrees smaller than the size specified for equal-spread subsampling. This procedure was repeated until all subtrees were below a target 'ceiling' size chosen to reflect the sizes of continental regions. We found that a ceiling size of 13,000 km summed MST length was most effective at dividing the global MST into natural geographic regions. Any remaining branches that crossed biogeographic barriers were manually removed and subtrees below the target size dropped.

From each of these subtrees, we drew 20 replicate subsamples of fossil localities, each having approximately equal summed MST length. This was achieved by progressively growing each spatial sample from a random starting locality until the target MST length had been achieved. The MST length chosen for standardizing spread must be large enough that each spatial sample contains sufficient occurrences to enable SQS subsampling at acceptable quorum levels. However, larger spreads tend to be more spatially uneven with respect to the distribution of localities within samples (for example, aggregation or discontinuities), and many important fossil-bearing regions have palaeogeographic spreads that may not meet larger target sizes. Through experimentation, we determined that spreads between 2,500–4,000 km strike an appropriate balance (see examples in Fig. 1). We chose to specifically use ∼3,200 km (±10%), which allows subsampled diversity estimates for North America in bins Tr4, J6, K7, K8, Pg1 and Pg2. Other sizes did not always return subsampled diversity estimates for all informative regions within Tr4, J6 and Pg1. J6 is an important time interval that includes the diverse and well-sampled fauna of the Morrison Formation, but palaeogeographic spread (∼3,400 km) is relatively limited, and as a result the data point disappears when the target spread is increased. Conversely, it is necessary to raise the target spread above 3,100 km to obtain estimates for the North America Tr4 and Pg1 samples.

Because of the nature of fossil locality data, which consists of discrete, often unevenly distributed spatial points, it is not possible to achieve perfectly uniform spatial subsamples. Adding a single locality may cause a subsample to overshoot the target size; in these cases, our algorithm accepted the subsample of localities (with/without the last-added locality) that fell closest to our target spread size. Our objective was not to achieve perfectly uniform subsamples, but reduce the variance of palaeogeographic spread among regions and time intervals. In this regard, we succeeded not only according to the summed MST length metric, but also according to alternative metrics (Supplementary Fig. 6 and Supplementary Table 2). Variation in the extent of unstandardized spatial sampling among continental regions and time intervals is substantial (CV of summed MST length is ∼80% for Mesozoic early Palaeogene non-flying terrestrial tetrapods; where more than one locality is present, individual data points range from 88–16254 km). The greatest reduction in variance is for summed MST length (over a six-fold reduction in the coefficient of variation), but convex-hull area and maximum GCD also see a threefold reduction in variance.

For comparative purposes, we also calculated subsampled diversity estimates for unstandardized spatial samples representing five separate continental regions (North America, South America, Europe, Asia and Africa; regions defined in Supplementary Table 1). Poisson regressions (GLMs using a log-link function) of diversity as a function of time were performed for the Mesozoic (bins Tr1-K8). Because our models use the canonical log-link function (appropriate for count data with a Poisson error distribution), these are log-linear regressions that model the relationship between diversity and time as an exponential function, the slope of which is an estimate of the net diversification rate (inverted because time counts down towards the present). We did not explicitly fit logistic models to the data because these are only appropriate for higher-resolution time-series data in which multiple data points provide evidence of both the increasing phase of diversity, and its subsequent static or equilibrial phase. However, because diversity-dependent models imply an initial rising phase followed by essentially static diversity, models were also run for the Mesozoic excluding bins Tr1–3. Richness estimates are only reported for data points associated with more than 20 references, which establishes a minimum threshold for worker effort or sampling intensity in an interval or region.

**Data availability.** All data and analysis scripts are available on FigShare (DOI: 10.6084/m9.figshare.4753711).

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

## Acknowledgements

We thank John Alroy for discussion, and all contributors to the Paleobiology Database. This research was funded by the European Union's Horizon 2020 research and innovation programme 2014–2018 under grant agreement 637483 (ERC Starting Grant TERRA to R.J.B.). P.U. wishes to thank The Leverhulme Trust for supporting his research via Grant RPG-129.

## Author contributions

R.A.C., R.B.J.B. and R.J.B. conceived the study. R.B.J.B. and R.J.B. contributed to the data set. R.A.C. analysed the data. R.A.C., R.J.B., R.B.J.B. and P.U. wrote the manuscript.

## Additional information

**Competing interests:** The authors declare no competing financial interests.

