## [Peer Review File · Nature Communications]

Reviewers' comments:

Reviewer #1 (Remarks to the Author):

I recommend major revisions for the manuscript, Controlling for the species-area effect supports constrained long-term Mesozoic terrestrial vertebrate diversification (NCOMMS-16-21330) before it is ready for publication. The main goal of the manuscript was to correct for heterogeneous spatial sampling in the reconstruction of Mesozoic terrestrial vertebrate diversity. Once heterogeneous sampling had been accounted for, the authors set out to determine whether the diversification trajectory was constrained (i.e., followed a logistic growth trajectory) or unconstrained (i.e., followed an exponential trajectory) [lines 45-60]. The final conclusion was that diversification was constrained through the Mesozoic. However, I am not convinced that this conclusion is fully supported by the analyses. I outline my skepticism below.

1) The minimum spacing tree (MST) subsampling metric is presented as the solution to the problem of spatial heterogeneity. However, there is no analysis presented that shows this metric performs better than any other metric, nor is there any citation for where this metric was developed or used elsewhere. In essence, MST's superiority is asserted without any evidence. More importantly, I'm not convinced that MST subsampling actually solves the problem of heterogeneous spatial sampling. Let's consider a hypothetical situation where there is a cluster of 100 collections in Alberta, Canada and second cluster of 10 collections in southern Utah (presumably if both regions had an equal number of collections there would not be a problem). If we then start randomly drawing collections until we get an MST of 3,200 km, won't the spatial distributions of the subsamples still reflect uneven spatial sampling? Moreover, let's consider a third cluster of 2,000 collections in, say, the Paris Basin. Let's also say that this third cluster covers the same area as the Alberta cluster above. Now we have a station where an MST of 3,200 km can be constructed in a single densely-sampled region, while in North America, the 3,200 km MST must span a much greater areal extent because that MST must connect collections in Utah and Alberta in order to reach 3,200 km. Again, it's unclear how these hypothetical North American and French MST subsamples are spatially equivalent.

2) The goal of the paper is to determine the trajectory of global tetrapod diversity during the Mesozoic. However, the MST subsampling method does not produce an estimate of global diversity. As pointed out on line 255, the diversity estimate is for an "arbitrary point on the species-area curve". Really what's been demonstrated is that there is relatively constant diversity at a given length scale (though see above that I'm not sure equal MST is the same as equal geographic coverage); answering the question of global diversification requires some estimate of turnover (or "beta") diversity.

3) The conclusion was drawn that the Mesozoic diversity trajectory was constrained. This was based on zero or weak slopes of diversity through time using a poisson regression (i.e., glm with log-link function). It seems to me that a model selection approach would be better. Why not compare a logistic function to a poisson or exponential function and see which fits the data better?

Minor Comments

The colored lines on figure 1 don't correspond to the colors on the color legend.

I have normal color vision, and I find it difficult to distinguish them in figures 2-5. Please use a combination of symbols and colors.

There is no citation given or methodology for calculation of the continental land area plotted on figure 5.

Given the number of correlations presented in Table 1, there should be some sort of p-value correction applied. Bonferroni is common, but there are other more finessed methods.

The PaleoDB collections were grouped by 1° grid cells, which are not equal-area. What impact does this have on the results?

It would be helpful to have a plot or table or some sort of data showing the severity of spatially heterogeneous sampling.

Lines 213-215 provide statistics on correlations between different metrics used and the raw data. It seems a bit absurd to me to call a Pearson correlation coefficient of 0.86 "weak". Especially when the made relative to coefficients of 0.84 and 0.91. I think most people would consider these all quite strong correlations.

Suggestions

An analysis showing how samples are distributed would be helpful? What is the null expectation for how samples should be distributed if there were no bias and how do the actual distributions differ? Are samples more clustered than expected? More dispersed?

Drop the section on the latitudinal diversity gradient. 1) Figure 4 is plotting diversity estimates of diversity for sub-continent size regions for different time intervals. The latitudinal diversity gradient is strictly a spatial pattern so plotting regressing a line through diversity estimates from different time intervals is nonsensical. 2) What is the latitude of a subsample with an MST of 3,200 km? That distance spans nearly 30°, so depending on the overall orientation of the samples, that could potentially be mixing collections from a huge range of latitudes. 3) There is conceivably a latitudinal diversity gradient story in these data, but I think that would be better explored in a separate manuscript.

Reviewer #2 (Remarks to the Author):

This MS makes an important contribution to the science of paleobiology, and I can find

nothing wrong with either the research approach or interpretation of the results. The writing is also excellent, with both clear and concise phrasing. I have only a few typographical errors to correct and minor questions for the authors, but otherwise gladly recommend this MS for publication.

Line 111 is missing a closing parenthesis after the citations.

Line 166 is missing an 'e' from 'been'.

Line 287 needs no comma, because it has a compound predicate.

Line 292: Why use "slope" diversity and not "beta" diversity?

Line 358: Please add a citation for the Scotese palaeomaps.

Reviewer #1 outlines his criticisms:

I recommend major revisions for the manuscript, Controlling for the species-area effect supports constrained long-term Mesozoic terrestrial vertebrate diversification (NCOMMS-16-21330) before it is ready for publication. The main goal of the manuscript was to correct for heterogeneous spatial sampling in the reconstruction of Mesozoic terrestrial vertebrate diversity. Once heterogeneous sampling had been accounted for, the authors set out to determine whether the diversification trajectory was constrained (i.e., followed a logistic growth trajectory) or unconstrained (i.e., followed an exponential trajectory) [lines 45-60]. The final conclusion was that diversification was constrained through the Mesozoic. However, I am not convinced that this conclusion is fully supported by the analyses. I outline my skepticism below.

1) Spatial sampling metric

Reviewer #1 writes:

1) The minimum spacing [sic] tree (MST) subsampling metric is presented as the solution to the problem of spatial heterogeneity. However, there is no analysis presented that shows this metric performs better than any other metric, nor is there any citation for where this metric was developed or used elsewhere. In essence, MST's superiority is asserted without any evidence. (emphasis added). More importantly, I'm not convinced that MST subsampling actually solves the problem of heterogeneous spatial sampling. Let's consider a hypothetical situation where there is a cluster of 100 collections in Alberta, Canada and second cluster of 10 collections in southern Utah (presumably if both regions had an equal number of collections there would not be a problem). If we then start randomly drawing collections until we get an MST of 3,200 km, won't the spatial distributions of the subsamples still reflect uneven spatial sampling? Moreover, let's consider a third cluster of 2,000 collections in, say, the Paris Basin. Let's also say that this third cluster has covers the same area as the Alberta cluster above. Now we have a station where an MST of 3,200 km can be constructed in a single densely-sampled region, while in North America, the 3,200 km MST must span a much greater areal extent because that MST must connect collections in Utah and Alberta into order to reach 3,200 km. Again, it's unclear how these hypothetical North American and French MST subsamples are spatially equivalent.

For reasons outlined below, it is not possible to demonstrate the general superiority of one measure of palaeogeographic spread over another. However, we have addressed this concern by 1) better explaining our use of summed minimum-spanning tree length (MST) for measuring palaeogeographic spread (see *Results, Methods, and Supplementary Methods*), 2) including appropriate citations to previous work using MSTs, and 3) performing additional analyses to demonstrate that this metric performs satisfactorily for our purposes and in fact represents a good compromise between other measures that capture various distinct aspects of spatial distributions. This mix of spatial signals is important, as we wish our metric to represent the most informative proxy for the size of the theoretical *geographic sampling universe*—and this plays a major role in dictating the size of the underlying taxon pool available for estimating diversity in deep time. Details are explained in the new portions of the text.

Please note that fossil localities represent point-pattern data, which may be spatially aggregated—to varying degrees—over a wide range of spatial scales. This makes the task of measuring their spatial distribution more challenging than it may be for some types of data for extant species. Our goal here was to reduce the variance in spatial spread among sets of samples and investigate the effect of doing so on variance in diversity through time and among regions. Arguably, this could be achieved by most adequate measures of spatial spread. Nevertheless, we think our analyses show that MST length is a good choice of method. Other researchers have rarely provided justification for the use of one metric over another, and to date no study has weighed their advantages and disadvantages or empirically evaluated differences in their performance. Therefore, as it currently stands, our MS is going well beyond the norm in terms of justifying and interrogating our choice of spatial metric.

Importantly, we have now performed a number of additional analyses that 1) establish how summed MST lengths relate to other, more commonly-used spatial sampling metrics; 2) demonstrate that our spatial subsampling procedures have substantially reduced the variability in geographic spread as quantified according to a range of spatial metrics; and 3) demonstrate that the reviewer's concerns about the heterogeneous distribution of localities within spatial subsamples are unfounded. These additional analyses are presented in the supplement. However, we also outline the key results of these additional analyses below:

1) We calculated palaeogeographic spread for each interval at regional level using a range of alternative metrics (convex-hull area, maximum GCD, counts of occupied 1° grid-cells, mean and median pairwise GCD and standard distance deviation). Pairwise bivariate plots (Supplementary Figure 2) show that summed MST length is tightly correlated with all other metrics, particularly convex-hull area (Pearson's $r = 0.91$), maximum GCD ($r = 0.97$) and standard distance deviation ($r = 0.91$). Moreover, summed MST length exhibits the tightest correlation with the number of occupied grid-cells ($r = 0.71$), demonstrating that the metric represents a good compromise between other metrics, and captures a combined signal of spatial coverage, dispersion and total extent. These strong correlations between metrics can also be seen in a time-series context in Supplementary Figure 3.

2) Our spatial subsampling procedure has substantially reduced variance in palaeogeographic spread between time intervals and geographic regions (Supplementary Figure 4; Supplementary Table 2). The greatest reduction in variance is for summed MST length (over a six-fold reduction in the coefficient of variation), but convex-hull area and maximum GCD also see a three-fold reduction in variance.

3) The reviewer was concerned that heterogeneity or clustering in the distribution of localities within spatial subsamples might bias our analyses. It is challenging to compare empirical distributions of spatial points to the null expectation for how samples should be distributed if there was no bias (i.e., if points were randomly distributed according to a spatial Poisson process) using standard point-pattern

statistics⁴ due to boundary issues and edge-effects. This is because spatial samples must be either overlain by quadrats and a value chosen for the total area within which sampling occurs, or use a nearest-neighbour index that is also extremely sensitive to the value chosen for total sampling area. To address this issue, we devised two metrics to quantify the degree of heterogeneity in the distribution of localities within spatial subsamples, a) the proportional contribution of the longest branch in the MST, and b) the coefficient of variation of the branch lengths within each MST. Plotting these metrics for each spatial subsample against time demonstrates that the level of intra-sample spatial heterogeneity are randomly distributed through time (Supplementary Figure 5). Furthermore, we note that standardising palaeogeographic spread according to other spatial metrics would be equally vulnerable to intra-sample spatial clustering of localities.

Lastly, we wish to acknowledge that although our additional analyses clearly support the use of summed MST length as a palaeogeographic spread metric, reviewer #1 made an important point about the potential for mismatch between MST length and other spatial metrics if the distribution of localities within a sample substantially differs. Binning localities within grid-cells considerably reduces these potential problems, as this procedure limits the contribution that large, densely-packed aggregations of localities can make to total spread. Furthermore, this argument against summed MST length implicitly rests on the primacy of some other spatial sampling metric (e.g. the total area described by a convex hull enveloping the outlying localities, or the maximum great-circle distance between pairs of localities). However, these purely range-based metrics are also redundant with respect to other important aspects of spatial sampling, such as the dispersion or coverage of localities within the study region (aspects that are equally important if we wish our spatial sampling metric to represent a meaningful proxy for the size of the geographic sampling universe). It would be very difficult, if not impossible, to standardise spatial samples of fossil localities with respect to all of these aspects, and we believe that summed MST length represents an appropriate compromise.

2) Type of diversity being measured

Reviewer #1 writes:

The goal of the paper is to determine the trajectory of global tetrapod diversity during the Mesozoic. However, the MST subsampling method does not produce an estimate of global diversity. As pointed out on line 255, the diversity estimate is for an “arbitrary point on the species-area curve”. Really what’s been demonstrated is that there is relatively constant diversity at a given length scale (though see above that I’m not sure equal MST is the same as equal geographic coverage); answering the question of global diversification requires some estimate of turnover (or “beta”) diversity.

Our paper does not seek to estimate patterns of global tetrapod diversity, and we do not claim this anywhere in the text. It is evident that there was some misunderstanding

here. So we have clarified our text at various points to ensure that it refers directly to regional diversity.

Although we draw our spatial samples from a global occurrence database, we are estimating diversity for standardised sub-continental or regional spatial scales. This is appropriate for our purposes, because diversity-dependence in the terrestrial realm is hypothesised to operate over regional or continental scales⁵. Although these standardised spatial samples are (as we point out in the manuscript) to some extent an ‘arbitrary point on the species-area curve’, they are at least the *same* point between time intervals and geographic regions, and indeed are the largest spatial scales that can be practically compared due to variation in palaeogeographic spread resulting from sampling biases.

It is clear from our work that, given the nature of the actual fossil record, it is not currently possible to produce a reliable global-scale diversity curve for tetrapods. We have added this point to the text of the Discussion. Variation in palaeogeographic spread among continental regions and time intervals is simply too great (continental regions often have no fossil record for many time intervals). Because of this dramatic variation in palaeogeographic spread, interpreting regional diversity estimates as true continental- or regional-scale gamma diversities is inappropriate. We have also revised the Introduction in order to emphasise that we are not seeking to estimate global diversity trajectories. However, there is likely to be some link between global and regional diversities; regional diversities may sum to global diversity in different ways as continents fragment. We have acknowledged this in the manuscript.

3) Appropriateness of statistical models

Reviewer #1 writes:

The conclusion was drawn that the Mesozoic diversity trajectory was constrained. This was based on zero or weak slopes of diversity through time using a poisson regression (i.e., glm with log-link function). It seems to me that a model selection approach would be better. Why not compare a logistic function to a poisson or exponential function and see which fits the data better?

The ‘Poisson’ component of Poisson regression concerns the error distribution of the response variable (and is appropriate for count data), not the form of the relationship between predictor and response. In fact, our general linear models *do* fit an exponential relationship between diversity and time: because of the log-link function, the response (diversity) is assumed to be an exponential function of the predictor (time). We aren’t trying to distinguish among models of diversification on short timescales. Instead, we’re trying to estimate the net diversification rate on a long timescale. GLMs with a log-link function are the most appropriate way to estimate the net diversification rate under a birth-death model⁶. We consider it to be inappropriate to fit a logistic model for the type of data we have: such a model would only be appropriate for much higher-resolution time-series data originating from a single geographic region and providing multiple data points on the initial rising phase of

diversity. In fact, it seems that the rising phase is short compared to the time granularity of our study. In any case, our spatially-standardised Mesozoic tetrapod diversity estimates can be seen to describe a near-flat line, with little variance. Because variance is low, we have little to gain by asking what model describes that variance using model-ranking. The key point, in fact, is that little change occurred of any sort.

Minor comments from Reviewer #1:

The colored lines on figure 1 don't correspond to the colors on the color legend.

Legend referred to heatmap scale representing sampling intensity at each palaeocoordinate; it has been removed.

I have normal color vision, and I find it difficult to distinguish them in figures 2-5. Please use a combination of symbols and colors.

We now use a combination of symbols and colours.

There is no citation given or methodology for calculation of the continental land area plotted on figure 5.

Continental land areas were derived from Smith et al. (1994) *Atlas of Mesozoic and Cenozoic Coastlines*. The citation has been added to the figure caption.

Given the number of correlations presented in Table 1, there should be some sort of p-value correction applied. Bonferroni is common, but there are other more finessed methods.

We disagree that corrections for multiple comparisons are appropriate, as they are widely recognised to reduce statistical power (e.g. ref⁷, cited in manuscript). As a clear illustration of this: the same analysis of a single dataset could be found as non-significant in a paper that reports multiple analyses, but significant in a different paper that reports just a single analysis. This clearly only makes sense when a 'scattershot' exploratory approach is taken to finding correlation between large sets of variables, which is not being done here. Furthermore, applying this correction is not conservative in the context of our study, which tests the hypothesis that standardising spatial variance reduces variance among data points (applying the Bonferroni correction makes it harder to detect change through time, and therefore provides potentially spurious support for this hypothesis). Nevertheless, we have added the Bonferroni-adjusted p-values for the GLM slopes to Table 1, and explained that we do not think this is appropriate in the text.

The PaleoDB collections were grouped by 1° grid cells, which are not equal-area. What impact does this have on the results?

'Equal-area' latitude/longitude degree-cell gridding has been used in many studies (e.g. refs^{2,8-11}). Because the vast majority of occurrences derive from similar palaeotemperate latitudes, our 1° grid cells have almost entirely uniform sizes, and the

differences among them are small compared to the absolute sizes of spatial regions. This is unlikely to have a noticeable impact on our results. We have explained this further in the Methods section of the MS.

It would be helpful to have a plot or table or some sort of data showing the severity of spatially heterogeneous sampling.

We have responded to this point in the section on minimum-spanning trees; see also Supplementary Figure 5 (and also see Fig. 5). There is no temporal trend in intra-sample spatial heterogeneity that might bias our diversity curves.

Lines 213-215 provide statistics on correlations between different metrics used and the raw data. It seems a bit absurd to me to call a Pearson correlation coefficient of 0.86 “weak”. Especially when the made relative to coefficients of 0.84 and 0.91. I think most people would consider these all quite strong correlations.

We have amended the text here.

Response to Reviewer #1’s ‘Suggestions’

An analysis showing how samples are distributed would be helpful? What is the null expectation for how samples should be distributed if there were no bias and how do the actual distributions differ? Are samples more clustered than expected? More dispersed?

See last paragraph of response regarding MSTs and palaeogeographic spread metrics. The situation is clear in Fig. 1 that shows some real palaeomaps and the clustering of localities. In a qualitative sense, the key point is that some spatial regions have zero localities, whereas other (in sedimentary basins) have lots. Similarly, some time intervals have much greater spatial sampling than other (see Fig. 5).

Drop the section on the latitudinal diversity gradient. 1) Figure 4 is plotting diversity estimates of diversity for sub-continent size regions for different time intervals. The latitudinal diversity gradient is strictly a spatial pattern so plotting regressing a line through diversity estimates from different time intervals is nonsensical. 2) What is the latitude of a subsample with an MST of 3,200 km? That distance spans nearly 30°, so depending on the overall orientation of the samples, that could potentially be mixing collections from a huge range of latitudes. 3) There is conceivably a latitudinal diversity gradient story in these data, but I think that would be better explored in a separate manuscript.

Broadly, we agree with the reviewer about the shortcomings of our original presentation of this information. We have chosen to clarify and improve our explanations, and Fig. 5, rather than omitting the point altogether. Addressing the referee’s point (1): We have also simplified Fig. 5 so that it more clearly emphasises latitudinal differences within specific time bins. Addressing the referee’s point (2): 3,200 km would span 30° if all samples were arranged in a line, but this is never the case (see Fig. 1). We have added the interquartile ranges of the spatial regions to Fig. 5.

Response to Reviewer 2

This MS makes an important contribution to the science of paleobiology, and I can find nothing wrong with either the research approach or interpretation of the results. The writing is also excellent, with both clear and concise phrasing. I have only a few typographical errors to correct and minor questions for the authors, but otherwise gladly recommend this MS for publication.

Line 111 is missing a closing parenthesis after the citations.

Line 166 is missing an 'e' from 'been'.

Line 287 needs no comma, because it has a compound predicate.

Line 292: Why use "slope" diversity and not "beta" diversity?

Line 358: Please add a citation for the Scotese palaeomaps.

Reviewer #2 strongly endorsed the paper, and we followed all of their minor suggestions verbatim:

Line 111: added missing closing parenthesis after the citations.

Line 166: missing 'e' added to 'been'.

Line 287: comma removed.

Line 292: Changed 'slope' to 'beta'.

Line 358: Citation added for the Scotese palaeomaps.

References

1. Gower, J. C. & Ross, G. Minimum spanning trees and single linkage cluster analysis. *Applied Statistics* **18**, 54 (1969).
2. Alroy, J. Geographical, environmental and intrinsic biotic controls on Phanerozoic marine diversification. *Palaeontology* **53**, 1211–1235 (2010).
3. Benson, R. B. J. *et al.* Near-Stasis in the Long-Term Diversification of Mesozoic Tetrapods. *PLoS Biol.* **14**, e1002359 (2016).
4. Burt, J. E., Barber, G. M. & Rigby, D. L. *Elementary Statistics for Geographers*. (Guilford Press, 2009).
5. Rabosky, D. L. & Hurlbert, A. H. Species Richness at Continental Scales Is Dominated by Ecological Limits. *Am. Nat.* **185**, 572–583 (2015).
6. Raup, D. M. Mathematical models of cladogenesis. *Paleobiology* **11**, 42–52 (1985).
7. Nakagawa, S. A farewell to Bonferroni: the problems of low statistical power and publication bias. *Behavioral Ecology* **15**, 1044–1045 (2004).
8. Alroy, J. *et al.* Phanerozoic trends in the global diversity of marine

- invertebrates. *Science* **321**, 97–100 (2008).
9. Alroy, J. The shifting balance of diversity among major marine animal groups. *Science* **329**, 1191–1194 (2010).
 10. Lloyd, G. T. & Friedman, M. A survey of palaeontological sampling biases in fishes based on the Phanerozoic record of Great Britain. *Palaeogeogr. Palaeoclimatol. Palaeoecol.* **372**, 5–17 (2013).
 11. Miller, A. I., Aberhan, M., Buick, D. P. & Bulinski, K. V. Phanerozoic trends in the global geographic disparity of marine biotas. *Paleobiology* **35**, 612–630 (2009).

REVIEWERS' COMMENTS:

Reviewer #1 (Remarks to the Author):

I have no further comments.